# An introduction to DUIA: The database on urban inequality and amenities

**Frederico Roman Ramos** [1,2]*, **Justus Uitermark**[1]

**1** Department of Human Geography, Planning and International Development –GPIO, Faculty of Social and Behavioural Sciences, University of Amsterdam, Amsterdam, The Netherlands, **2** Department of Planning and Economic Analysis Applied to Administration–PAE, São School of Business Administration FGV-EAESP, São Paulo, Brazil

* f.ramosroman@uva.nl

**Data Availability Statement:** The original data underlying the results presented in the study are available from (IPUMS International https://international.ipums.org/international/). The

## Abstract

This article introduces DUIA, the Database on Urban Inequality and Amenities. DUIA includes data on the socio-economic development and amenities of 86 cities. The database especially covers cities outside the West, providing new opportunities for comparative research in fields suffering from a dearth of data. DUIA addresses three concerns that have not been resolved by other databases on cities. First, we draw upon remote sensing derived data from the Atlas of Urban Expansion to more accurately define city boundaries. Second, we draw upon survey data stored in IPUMS (Integrated Public Use Microdata Series) to include extensive, harmonized, and disaggregated data. Third, we use open source software and share our scripts to ensure transparency and replicability. DUIA includes information on dwelling and household characteristics, educational attainment, ownership of assets and appliances, and access to amenities. We provide illustrative analyses on asset inequality and water access to demonstrate the potential for the database. Although we also identify several limitations, DUIA represents a step forward in the systematic study of inequality and amenities over time and across cities.

## Introduction

As methods for data acquisition and processing advance, opportunities to do comparative research with a large sample of cities increase. Although recent years have seen a proliferation of databases with city-level data, there is still a long way to go before we achieve a comprehensive and authoritative data infrastructure. It is particularly challenging to construct databases with broad international coverage since the quantity of data available and ways of measurements differ sharply between countries. As a result, databases typically include only a limited number of cities in the Global North or a limited number of variables. Perhaps the most ambitious attempt up to date to create a database with comprehensive coverage is the Global Urban Indicators Database produced by UN-Habitat.

Although presently available databases provide an important and a rich source of information, they also suffer from shortcomings. First, urban databases generally assemble data from different sources, which means that data are generated according to different methods and are

aggregated city level data is available at https://github.com/fred-r-ramos/DUIA.

**Funding:** J.U. #452-17-003 Netherlands Organization for Scientific Research The funders had no role in study design, data collection and analysis, decision to publish, or preparation of the manuscript.

**Competing interests:** The authors have declared that no competing interests exist.

not fully harmonized. For instance, governments might have different ways of gauging access to improved water, which means that differences might reflect administrative procedures rather than facts on the ground. The Global Urban Indicators Database [1], the most ambitious database with coverage in the so called Global South, is a case in point as its online dashboard allows users navigate tables with unharmonized variables. Second, existing databases typically provide city-level indicators, which makes it impossible to disaggregate data, which, inter alia, severely limits the possibilities to study inequalities. For instance, while Global Urban Indicators Database does provide an estimation of the proportion of the population with access to improved water, we would not know which segments of the population have access to improved water. This is the case for most of the database initiatives, including those composed by governmental and civil society panels [2–4] and also for initiatives that seeks to compare cities according to its economies of scale and market size [5–8]. Third, besides some exceptions that prove the rule [9, 10], existing databases often do not provide access to underlying data and procedures making it impossible to replicate analyses.

This article introduces DUIA, the Database on Urban Inequality and Amenities, which goes some way toward addressing these issues. DUIA includes data on the socio-economic development and amenities of 86 cities from a total of 32 countries. DUIA comes with an integration protocol and code in R scripts, making both the construction of the database as a whole and specific statistical analyses fully transparent and replicable. Three considerations guided the construction of DUIA. First, we draw upon remote sensing derived data from the Atlas of Urban Expansion to define city boundaries as accurately and consistently as possible across the different countries. Second, we draw upon survey data stored in IPUMS [11] to include extensive, harmonized, and disaggregated data. Third, as we especially seek to contribute to comparative research outside the West, we developed tailor-made solutions to include Indian and Chinese cities for which data were not available in IPUMS.

The remainder of this paper is structured as follows. The following section presents an inventory of city-level databases and briefly discusses their objectives and approaches. The third section presents the procedures adopted in DUIA. In section four, we develop some illustrative examples to demonstrate the functionalities of the database: we look at water to illustrate the potential for studying access to amenities and we look at assets to illustrate the potential for studying inequalities. We discuss DUIA's limitations as well as some opportunities in the fifth section before offering our conclusion in the sixth section.

## Urban databases initiatives

In recent years various organizations have been developing comprehensive urban databases, including Africapolis, the Global Power City Index, the Brookings Global Metro Monitor, the PWC Cities of Opportunity, the Global Cities Lab Top 500, the GaWC City Classification, the JLL's City Momentum Index, the *Economist*'s Global Livability Index, the C40 Compact of Mayors, the Metropolis Observatory, and the Urban Age Project (Table in S1 Appendix). This informational landscape gives us an unprecedented capacity to compare cities through different lenses and for different purposes. However, as the variety and complexity of the information increases, more attention to transparency in definitions and methods is necessary, especially when the goal is to include cities in countries where statistical information is less readily available. For instance, the World Council on City Data address this issue by developing standardized urban metrics, including the creation of an ISO standard-setting.

The most ambitious attempt to track the socio-economic development of cities across the world is probably UN-Habitat's New Urban Agenda [12]. From 2003 onwards, UN-Habitat has pioneered efforts to improve the availability and reliability of urban indicators. Currently,

UN-Habitat's Global Urban Indicators Database covers 1500 cities in 132 countries with 77 indicators (as accessed in 23/05/2021 at https://urban-data-guo-un-habitat.hub.arcgis.com) many of them related to tracking progress with respect to the Sustainable Development Goals (SDG). UN-Habitat makes the data freely available and provides extensive documentation. Another important quality is its sampling procedure. The basic idea behind the database is to sample 1500 cities that are representative of all cities of more than 100,000 residents in 2010, though it should be noted that the website does not give exact information on which cities were included nor does it provide aggregate statistics for the sample as a whole. Africapolis is another valuable initiative. By using remote sensing, it provides a standardized procedure for the identification and delineation of cities and towns. The Worldpop Initiative, too, uses satellite imagery to identify and delineate settlements. It combines a range of geospatial datasets to cover a wider range of variables and, importantly, is open access, providing greater opportunities for both collaboration and corroboration.

All databases of urban indicators face a large number of challenges in collecting reliable and comprehensive data. Here we focus on three key challenges that we have sought to resolve. The first is about the conceptual definition of "the city" and the operationalization delineation of cities. National governments have very different administrative practices of defining and delineating cities, which stands in the way of having a general definition of what constitutes a city and a general strategy of delineation which areas belong to cities. The second concerns the comprehensiveness and diversity of cities. The scope of analysis will depend on the number and variety of cities that are represented in the database but adding more cities and countries typically means that researchers have to reckon with more unharmonized data. The final challenge concerns the transparency and replicability of the procedures for collecting and processing data as well as the availability of data and code. This involves the variable selection, data extraction, data compilation, and analytical outputs such as graphs and tables. The following subsections review how urban databases have addressed the challenges and explain the rationale behind DUIA.

## Definition of cities

Most urban databases rely on administrative definitions of cities. This is problematic since administrative categories for defining and delineating cities differ greatly across countries. In addition, there is often a mismatch between the cities as administrative units and cities as functional entities. This issue has been debated extensively within the context of monitoring progress on SDGs. From 2018 onwards, UN-Habitat has been organizing a series of workshops (reports on country consultations towards a harmonized urban and rural areas definition available at https://urban-data-guo-un-habitat.hub.arcgis.com/pages/stories as accessed in accessed on 23/05/2021 with representatives of national statistical offices to discuss the possibilities for developing a global definition of cities and harmonized criteria to identify urban and rural areas. One promising avenue is to use remote sensing and gridded population data to depict urban agglomerations [13]. The major advantage of this data source is that it allows the identification of land-cover patterns in a consistent way with a planetary coverage. Leyk et al. [14] present a detailed comparison among these initiatives. Other studies use auxiliary data to indicate urbanized areas such as density and commuting flows [15]. Some urban databases, including Africapolis, incorporate remote sensing data to chart urban sprawl and land-use patterns.

We build on these efforts by defining using the definition and operationalization of cities in the Atlas of Urban Expansion (AUE) [16]. The objective of the AUE is to monitor urban expansion by mapping and measuring key attributes of urban expansion in three time periods

through the classification and analysis of Landsat satellite imagery. AUE uses digital image processing techniques to identify the city built-up area extension, its *extrema tectorum*. For each city, the AUE freely provides two types of data: first, an aggregated dataset with urban morphological metrics, patterns of urban expansion, and some estimations of population size and density extracted from demographic datasets; second, georeferenced layers containing the city boundary and urban footprint as well as a collection of raster layers that resulted from the image classification process. As we explain below, we use the morphological data and georeferenced layers to match census data as closely as possible to urban areas. In brief, the matching process is possible since each household microdata sample in IPUMS can be associated to a variable that indicates the code of the second-level Enumeration Areas (EAs) where it is located. As we elaborate below, we use the georeferenced layers provided by AUE and IPUMS to select the EAs that are completely or partially covered by the Study Area defined by the AUE.

## Comprehensiveness and diversity of cities

The criteria for including cities in databases are often implicit or vague (like "the world's largest cities"). Here, again, UN-Habitat has been leading the way by proposing a method to define a representative Global Sample of Cities for the sake of comparative analysis. The method was developed in a joint initiative with the Lincoln Institute for Land Policy and New York University and has also been adopted by the AUE. The UN Sample of Cities includes 200 cities which represent 5% of the universe of 4,231 cities of over 100,000 inhabitants in 2010, comprising 70% of the world's urban population. The sample is constructed through a threefold stratification procedure that ensures that cities of all sizes, from all regions, and from large and small countries are represented. In the first step, the cities were sampled at random from eight world regions in proportion to the urban population in each region (the regions are East Asia and the Pacific; Southeast Asia; South and Central Asia; Western Asia and North Africa; Sub-Saharan Africa; Latin America and the Caribbean; Europe and Japan; and Land-Rich Developed Countries). In the second step, an approximately equal number of cities were selected at random from four ranges of population size, each range containing one-quarter of the total population of the cities in the universe (the population size groups are 100,000–427,000; 427,001–1,570,000; 1,570,001–5,715,000; and 5,715,001 and above). Finally, the third step observed three country groups identified by the number of cities in the country in proportion to the urban population in each group (Countries with 1–9 cities; 10–19 cities; and 20 or more cities). For DUIA, we adopt this same sample as a starting point for data compilation.

## Meta-data and replication

One of the major challenges for building urban databases is to include reliable information on relevant variables. There is typically a tradeoff between, on the one hand, using standardized procedures and, on the other hand, including a wealth of information. Africapolis and the AUE use standardized procedures to identify and delineate cities but do not include a wealth of information. The UN-Habitat database does have an array of variables but they were obtained from different sources and calculated in different ways. Despite some recent refinements in the data selection and visualization, the UN-Habitat database still lacks the capacity to replicate methodological procedures. As it is built from contributions from local urban observatories, it is almost inevitable that are significant differences in how data is processed and the availability of metadata. Information about primary data sources, formalization of calculations, and conceptual definitions are absent for several indexes and data tables. Navigating the informational content is also far from ideal since it has no clear categorization of themes of

interest. As an example, a search using the tag 'poverty' returned only one table on poverty rates at national level (the data catalog was accessed in 23/05/2021 using the TAG 'poverty' as follows https://data.unhabitat.org/search?Collection=Dataset&tags=poverty) which is, according to the metadata provided, based "on income per capita estimated from household survey data. . ." This type of information is insufficient for an authoritative comparative analysis since differences in the household surveys, variable definitions and methodological procedures are not transparent. Moreover, since we only have access to an aggregate score on a synthetic measure, it is impossible to conduct fine-grained analysis.

These issues suggest that databases like DUIA need to optimize possibilities for replication. As Christensen et al. [17] suggest, replication means *verification* whether the specific results of the original study can be reobserved. In a broader meaning, it also means *reanalysis*, or whether conclusions drawn from the analysis are robust. Recent technological advancements in open source software and public repositories are expanding the possibilities replication thus understood. The World Bank's *SDG Atlas* is a clear example of this trend. All the calculations, graphs and maps were constructed using open source software (R and QGIS) and all the code and data are openly available in a GitHub repository [10] providing full replicability and transparency for the calculations used in the database. Similarly, initiatives such as the WordPop gridded population [18] and the OECD functional urban areas definitions [13] also made their source code and data available, mainly using R scripts and GitHub repositories. DUIA followed this model: all the data gathering used open and freely available sources and all the methodological steps were developed using R scripts available in GitHub.

## Building a city-level database

### Data sources

DUIA relies on two freely and easily available main sources: the IPUMS-international (IPUMS originally stood for Integrated Public Use Microdata Series). All the data we use in our analysis are derived from IPUMS-International (for simplicity we refer to IPUMS-International as IPUMS) and the Atlas of Urban Expansion. Both data sources are maintained by renowned academic institutions: the Institute for Social Research and Data Innovation at the University of Minnesota is responsible for IPUMS. The AUE is a multi-institutional initiative led by the Marron Institute of Urban Management and the Stern School of Business of New York University and developed in partnership with UN-Habitat and the Lincoln Institute of Land Policy.

IPUMS in 2021 offers census data from 98 countries. IPUMS includes sample data, with sample densities ranging from 1 percent to 10 percent of national populations. Composed of microdata samples, IPUMS provides information on the individual and household level, enabling tabulations and cross-tabulations tailored for different research questions. This means, for instance, that we not only know which proportion of a city's population has access to water but also which groups lack or have access. Nationally representative samples are systematically drawn from the total enumerated population by IPUMS or national statistical offices [19] IPUMS adds extensive and integrated metadata. For our purposes, it is important that it provides GIS boundary files, making it possible to establish whether households belong to a city (as defined by the AUE; see below). Another crucial contribution by IPUMS to comparative research is the harmonization of variables, minimizing the difficulties that arise when questionnaires are not identical.

For some countries, the availability of microdata variables is still limited in IPUMS, precluding analysis at the individual or household level. These countries include China and India, two countries with large populations and many cities included in AUE's sample of cities. To

include these two countries in DUIA, we used aggregated city-level census variables obtained directly from the respective national statistical agencies.

For Indian cities, we accessed aggregated data for the censuses of 2001 and 2011. The Indian Census agency maintain a web page (https://censusindia.gov.in/DigitalLibrary) where it is possible to freely download the census table by township (*tehsil*). It also provides census base maps as images (pdf) containing the description and extension of each tehsil. To select the tehsils that best correspond to the city boundary in AUE, we georeferenced the pdf maps and used them as background image GIS layer to identify the list of Tehsil to be included in each city through an overlay procedure.

For China, we obtained aggregated values calculated on a 10% sample from the 2000 and 2010 census. The Chinese census provides data for different administrative subdivision levels. We use the prefecture level data to represent cities. DUIA uses case-specific code to acquire and preprocess data on Chinese and Indian cities. Data is then integrated only at city-level in the database, allowing the comparison with the other cities in the database. While we felt it is important to include Chinese and Indian cities, we note that we unfortunately cannot do analyses at the household and individual level. In addition, for Chinese cities, we could not use the urban extension defined by the AUE.

All in all, DUIA includes data on 56 cities obtained from IPUMS and of an additional 30 cities in China and India, bringing the total to 86 cities, distributed over five continents. The Fig 1 maps cities included in the database. Although this a large sample with cities from many different contexts, we acknowledge that it is not representative of any population of cities. This means, inter alia, that the aggregate statistics we report below are for descriptive purposes and not meant to generalize to a particular class of cities. The real value of the database resides in the opportunity to compare cities on a comparatively wide range of variables.

## Data selection and integration

The method to assemble DUIA involves three main steps: data selection and extraction; data compilation and integration; and aggregation of city-level statistics. All data, code, and software are freely available: each step can be replicated using the R scripts accessible on the

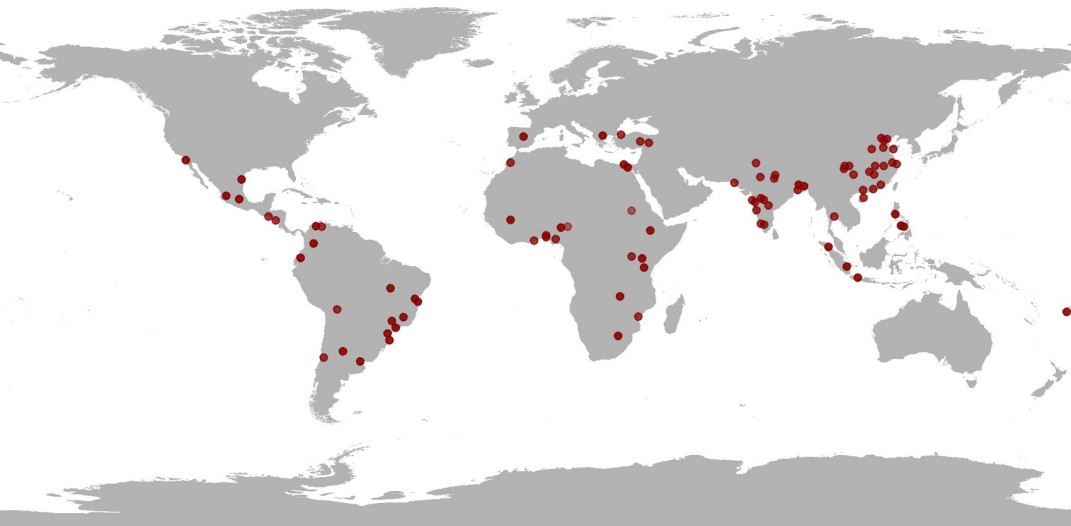

**Fig 1. Cities included in DUIA.**

GitHub of the project and the main data providers for DUIA have all the data freely available on interactive websites. The diagram in Fig 2 offers an overview of DUIA's workflow.

The first step consists of the data selection and extraction. We adopted the Sample of Cities as defined by the UN-Habitat and the AUE as the starting point to the sampling strategy. One key aspect of this sampling strategy is the definition of cities by their geographical extent, or "as agglomerations of contiguous built-up areas (and the open spaces in and around them) that may contain a large number of municipalities but, more often than not, constitute a single labor market." [16]. For each one of the 200 cities, AUE defines the extension of the built-up area in a study area in three periods of time, around the years 1990, 2000 and 2015.

To match the cities as defined through AUE to the IPUMS census data, we first verify whether census data is available for the countries that have cities included in the sample. This is the case for a total of 167 cities distributed over 57 countries. The data extraction in IPUMS

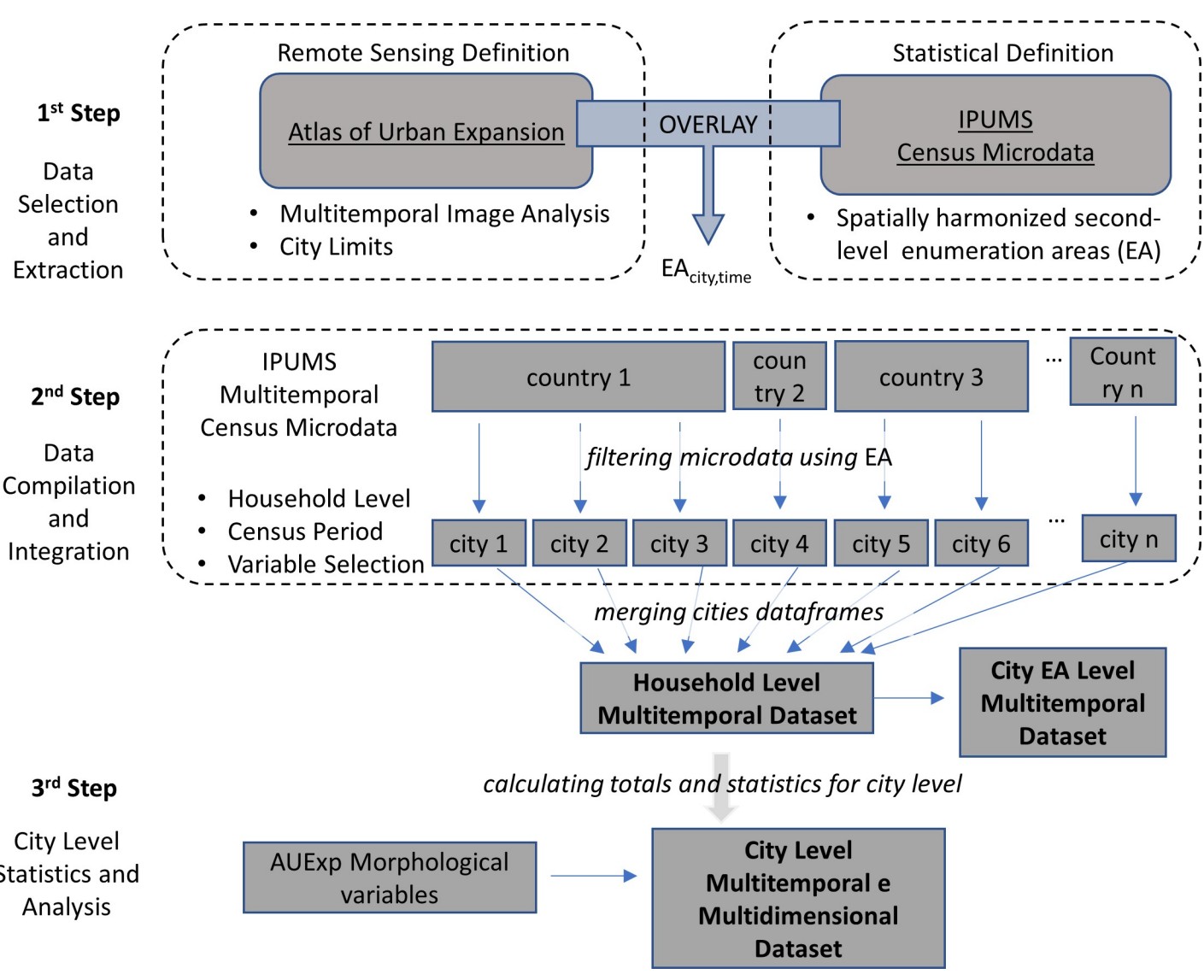

**Fig 2. Schematic DUIA's workflow.**

is done through a web-based interface that allows for a selection of samples and variables. For building DUIA, we focused on the variables that are related to the household characteristics including tenure, access to utilities (e.g. water, electricity, etc.), assets and appliances (e.g. autos, TV, etc.), and dwelling characteristics (e.g. building materials, number of rooms, etc.). There are substantial differences in terms of variable availability and spatial-temporal coverage. S2 Appendix shows the variables available for each country and period in IPUMS. After excluding cities for which very limited information was available, a total of 56 cities remained in the database.

Among the variables in IPUMS is a geographical code indicating the second-level subnational administrative unit in which the household was enumerated. To match the data from IPUMS and the AUE, we use an overlay procedure to see whether the second-level administrative unit is part of the city as defined by the AEU. This is straightforward when there is a perfect spatial match between the AUE study area boundaries and the second-level administrative units. S3 Appendix describes the matching for Dhaka and Mexico to explain how we treated cases where there was not a perfect spatial match. In addition to spatial matching, we need to do temporal matching. Fig 3 shows a comparison between the years of the satellite imagery used in the AUE and the year of the censuses available in IPUMS. Although there is no perfect fit, it is possible to group into three periods of analysis.

The second step in the workflow consists of the compilation of all the household samples available for each country and the selection of the samples that correspond to the second-level EA within the boundaries of the study area as defined by the AUE. This generates a series of household-level, multi-temporal sample data frames that are then merged into a unique repository, an integrated, household-level database. This database contains the information of the second-level administrative units for every household level sample, allowing some degree of

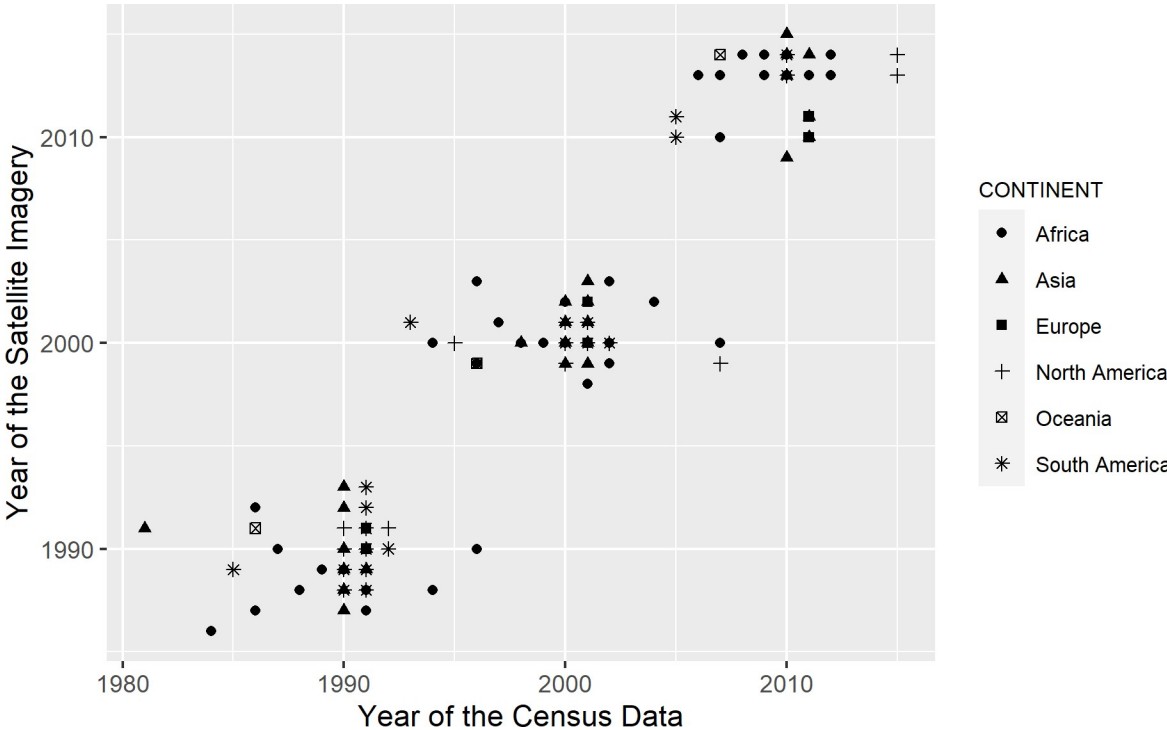

**Fig 3. Comparison between the year of IPUMS and AUE data acquisition.**

intraurban analysis for most of the cities. For instance, for Mexico City, we can compare across 60 districts, for Accra among 10. Cities in the sample can subdivided into 10.2 areas on average.

The final step comprises the calculation of the aggregated city-level statistics from the household samples. This generates a data frame with the city as the unit of observation, which can then be integrated with the variables on urban morphology derived from the AUE. It also allows the integration of aggregated city level information available from the census data as the case for Chinese and Indian cities.

## DUIA: Features and components

The outcome of the extraction and integration processes described in the previous sub-sections is an informational infrastructure consisting of (1) an aggregated city-level database publicly available via the figshare service, (2) a microdata household-level database maintained internally under access control, and (3) a collection of R codes and data extraction specifications available in a GitHub repository. The controlled access to the microdata is due to restrictions in the IPUMS policy for the use and dissemination of the microdata. However, the data extraction specification and the openly distributed R codes allow the full replication of the data extraction process and the household-level database assemblage.

The core of the DUIA is the microdata household-level database, a file in an R dataframe format (Rda) composed of approximately 50 million household samples that allows multivariate analysis and tailored case selections and aggregations. The openly available aggregated city-level database is compiled from the household-level database and includes all the selected harmonized variables for all the cities in different periods of time. The city-level database allows simple and direct access to 33 variables in different categories such as 'access to piped water', 'internet access' and 'educational attainment'. It is a ready-to-use spreadsheet and does not require the use of R codes. As it has the city as the unit of observation, it allows for the integration with the tables from the AUE.

The R codes stored in the GitHub repository are organized in folders according to the different stages described in the DUIA's workflow (Fig 2). The first folder contains a set of country-specific codes that enables the selection of the IPUMS samples based on the geographical overlay operation described above. It uses the year-specific second-level geography shapefiles available at the IPUMS webpage and the GIS databases retrieved from the AUE. The GitHub repository further includes a set of text files describing how country samples were extracted from IPUMS to enable replication. The second folder contains the code used in the integration process that has as outcome the microdata household-level database. It outlines the operations used to make the country-specific data compatible, including standardization of variables and formatting. The third folder contains the code used to generate the aggregated city-level database. It further includes the code used in the illustrative analyses hereafter presented. The repository is open for contributions, making it possible to incorporate new R code for processing or analyses.

## Illustrative analyses

The database allows for a wide range of analyses into various kinds of inequalities and access to different kinds of amenities. Where IPUMS microdata is available, there is a possibility to conduct research at the level of the household. This represents an important quality since many of the currently available databases only present aggregate indicators and indexes. The following subsection illustrates this potential through a study of the development of inequality over time and across cities by means of an asset inequality index. The second subsection

illustrates the potential of database for comparing differences in the access to amenities between and, to some extent, within cities through an analysis of differential access to piped water.

## Inequality in household assets

Rising inequality is one of the most pronounced and concerning trends in recent decades. There is ample evidence that, while inequalities between countries are declining as a result of fast economic growth in previously poor countries, inequalities within countries are on the rise. Cities both generate and reflect such rising inequality [20, 21] and inequality generally increases with city size [22, 23]. Marked differences in terms of wealth and income are especially pronounced in countries with weak welfare states. Some authors, like Davis [24], argue that while globally connected elites might escape poverty, the growing mass of urbanites has bleak prospects and will suffer impoverishment. However, there are also reasons why we should not expect rising inequality. Studies of informal neighborhoods have found that residents over time manage to drastically improve their material circumstances [25, 26].

DUIA offers various opportunities to study inequality within and between cities. Here we illustrate its potential through an analysis of inequalities with respect to household assets. Household assets here refer to resources for a leading a comfortable life, including ownership of durable consumer goods (like a fridge or a television) and residence in a good dwelling (as measured by e.g. the number of rooms or the material from which it is made) [27]. One reason to look at household assets is pragmatic: in the absence of abundant and comparable data on income and wealth, it is pragmatic to focus on household assets for which information can be obtained from census data. Another reason is analytical: household assets arguably are important for wellbeing, comfort, and dignity and sharp differences in this respect represent a tangible and consequential expression of inequality. We provide tentative answers to two questions: is inequality in terms of household assets growing or decreasing? And, is household asset inequality greater in cities than in the countries they are part of?

To construct our index for household asset inequality, we replicate McKenzie's approach but take cities, not countries, as objects of analysis [27]. The basic idea of the index is to use Principal Component Analysis (PCA) for reducing multidimensionality commonly present in large datasets and to measure inequality on the first component, that is, the component that accounts for most of the variance. The approach calculates a *relative* measure of inequality, using the first principal component in each period of interest relative to the first principal component in the sample in both periods. Formally, given asset vector $x_1$ the first principal component of the observations, $y_i$, is the linear combination whose sample variance is greatest among all such linear combinations. The inequality index for a period in time $t$ is defined as:

$$I_t = \frac{\sigma_t}{\sqrt{\lambda}} \tag{1}$$

where, $\sigma_t$ is the sample standard deviation of $y_i$ across households in period $t$, and $\lambda$ is both the eigenvalue corresponding to the first principal component, and also the variance of $y_i$ over the whole sample in all periods (as demonstrated by McKenzie [27] p.258). For this reason, the value of the index is relative to each city or country, not allowing interpretation of the absolute level of inequality between cities. It provides, however, a good and synthetic reading of the evolution of inequality in cities. Moreover, it is possible to compare local trends to national trends.

The household asset index is based on variables related to homeownership, dwelling amenities, building material, and ownership of consumer durables. The census data available in

IPUMS covers most of these dimensions but with differences among countries. On average, the dataset contains 15 variables covering household assets. We used as many variables as possible in our calculations and included only those cases where more than 10 indicators were available. All the variables we use are binary except for the number of rooms per person, which we calculated by dividing the household size by the numbers of room in their residence. The harmonization process applied by IPUMS makes it straightforward to use the variables but the binarization of many of the variables demanded *ad hoc* coding for each country. S2 Appendix provides an overview of the variables used for the calculation of the index for each country.

The graphs included in Fig 4 show the inequality index for 36 cities and their countries (in red). Most cities– 30 out of 36 –show a decrease in inequality over time; a remarkable finding in light of research documenting drivers and patterns of growing income and wealth inequality. Moreover, we find very little evidence that the development of cities is disconnected from their national context (cf. [21]): there is a correlation coefficient of 0.88 between the city-level and country-level inequality index, indicating that the development of city-level inequality

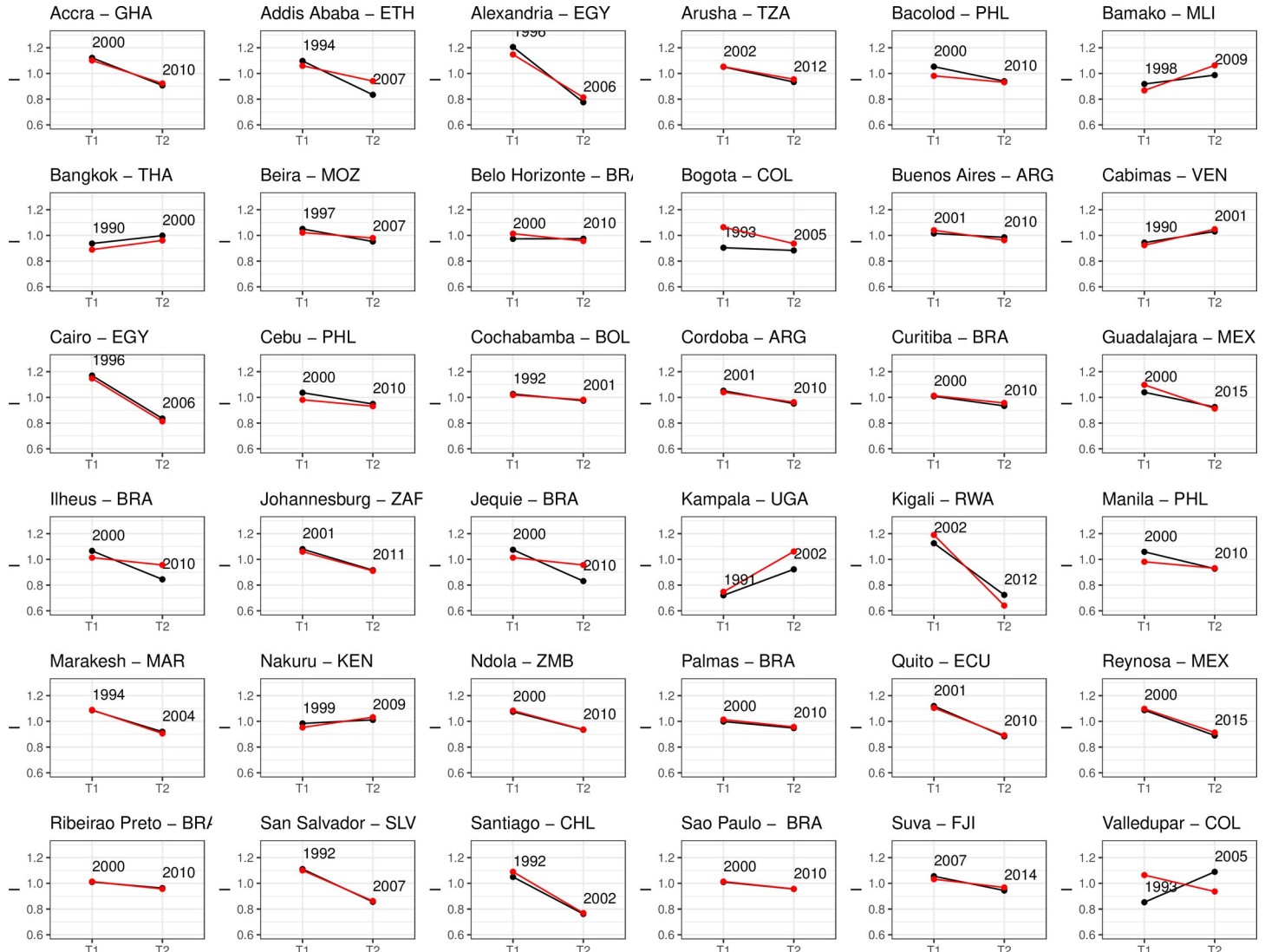

**Fig 4. Evolution of Asset Inequality Index for selected cities (in black) and countries (in red).**

tracks country-level inequality. The only exception is Valledupar, a city in the north of Colombia, where inequality increased despite the overall decrease observed in the country. To the extent that there are differences in the evolution of inequality between cities in the nations in which they are situated, the average annual variation rate of the asset index for all the cities in the sample is-1.1, and for the countries the average annual variation is slightly slower, -0.97.

On average, the decrease in inequality was most significant among African cities with an average annual rate variation of -1.6 percent per year, followed by the Mexican cities with -1.2 percent per year, South American cities with -1.1 percent per year, and Philippian cities with -0.8 percent per year (see Fig 5).

Fig 6 presents changes in the Asset Inequality Index over time. In our sample, the Ugandan capital Kampala shows the sharpest growth in inequality with an average increase of 2.3 percent per year, which is still lower than the increase in inequality in the country of Uganda as a whole (3.2 percent per year). The second highest rate was observed in the Columbian city of

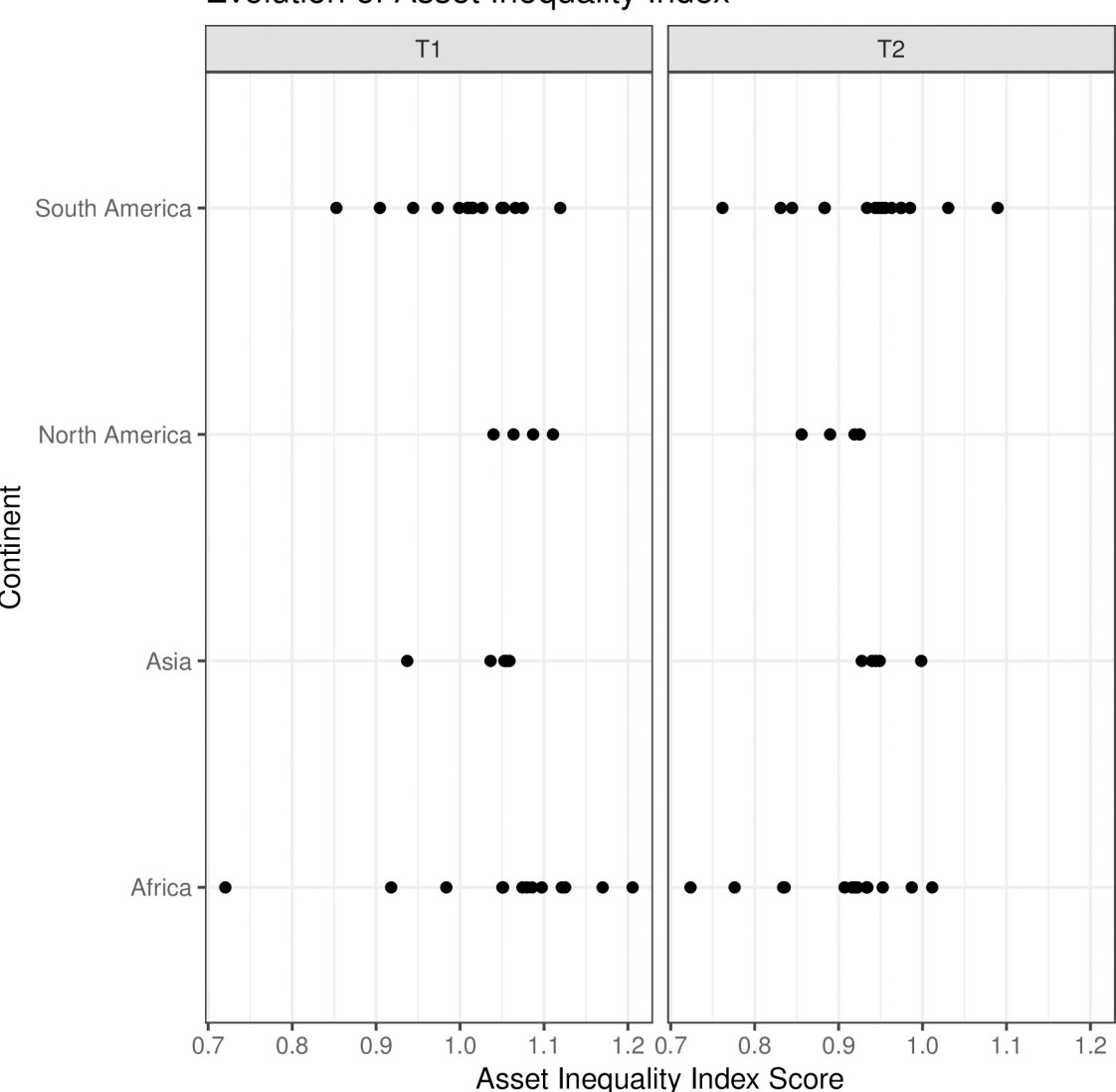

**Fig 5. The development of the Asset Inequality Index by continent.**

## Anual Evolution of Asset Inequality Index

**Fig 6. Annual variation of asset Inequality Index by city.**

Valledupar, with a growth in inequality of 2.1 percent per year, which contrasts with the trend toward equality at the national level, suggesting that there must be specific local factors at play. The group of cities with decreasing asset inequality– 80 percent of the sample–shows even more marked change. Both the Rwandan capital of Kigali and Alexandria in Egypt show annual decreases in the inequality index of 4 percent.

Although there are several cities that exhibit greater inequality, it is remarkable that our findings suggest that changes in the opposite direction–toward greater equality–predominate in most cities in our sample. Considering that cities are often seen as magnifiers of inequality, it is noteworthy that we find that trends for cities and countries are generally remarkably similar. Since we here discuss a specific kind of inequality (in terms of households' assets) for a specific sample of cities (those for which we have sufficient data), we do not intend to generalize from our findings. Nevertheless, our findings do minimally suggest that, at least in some domains and in some cases, there are trends toward equality that have received scant attention in the literature.

## Inequalities in access to piped water

Secure access to potable water is a life necessity but far from self-evident for a large proportion of the world's population. It is listed as one of the SDGs and has been receiving the attention of a vast amount of studies and analysis. DUIA offers a framework for studying how access to water develops over time and across different cities on the basis of census data. To illustrate its potential, we conduct an exploratory comparative analysis using IPUMS' harmonized variable for water provision (WATSUP), which "describes the physical means by which the housing unit receives its water. The primary distinction is whether or not the household had piped (running) water." (According to the metadata available at https://international.ipums.org/international-action/variables/WATSUP#description_section). Although IPUMS includes information to distinguish various kinds of water access, for the sake of comparing across a range of cities and over time we here distinguish two basic categories: 'piped water' and 'no piped water.' For the first period, we have data for 31 cities, for the second on 33 cities, and for the third on 26 cities. We aggregate the household microdata to the level of the city using the weighting variable (the weighting variable—HHWT or household weight—indicates the number of households in the population represented by the household in the sample). Since IPUMS data on water provision is not available for China and India, we included aggregate census data obtained directly from the respective official statistical agencies for 13 Indian and 21 Chinese cities.

The collection of graphs for the 86 cities presented in Fig 7 show the evolution at a glance. While there is substantial variation in trajectories and levels of piped water provision, there is a general trend in the direction of increasing access. On average, the cities in the sample had 74% coverage of piped water in the most recent period covered by the sample, representing an increase in 4 percentage points in comparison to the previous period.

Grouping the cities by continent (Fig 8), we see that Latin-American cities have steadfastly expanded access to piped water and reached almost universal coverage in the last period of analysis. African and Asian cities show much greater variation. Some cities achieve universal coverage, but others see access to piped water reduced. The drop in access for cities in the Philippines is noteworthy: Manila decreased from 73 percent coverage in 1990 to 57 percent in 2014 and Cebu from 50 percent to 33 percent in the same period.

So far, we illustrated the potential of DUIA for macroscopic analysis, providing a broad overview of developments across a wide range of cases. For such broad analyses it is necessary to reduce complexity, for instance, by transforming variables into dummies and ignoring variations at the sub-city level. When the analysis focuses on specific cities or countries, however, it is possible to conduct more fine-grained analysis.

## Limitations and opportunities

Although DUIA makes some important steps, the database also suffers from a number of limitations. First of all, there is a trade-off between, on the one hand, representativeness and, on the other hand, reliable, comparable, and granular data. Whereas UN-Habitat's sampling strategy for the Global Urban Indicators Database optimizes for representativeness, DUIA prioritizes reliable, comparable, and granular data. This choice means that the sample of cities in DUIA is not representative and can at best be used to approximate general trends. Second, the data in DUIA is as good as the censuses from which they are sourced. The censuses differ in terms of coverage, are conducted in different moments in time, and generally lack information on e.g. beliefs and values or income and expenditures. A third limitation is that DUIA requires considerable computational power since the data are processed from the census samples. For instance, the computation underlying the analysis of the household asset index for cities and

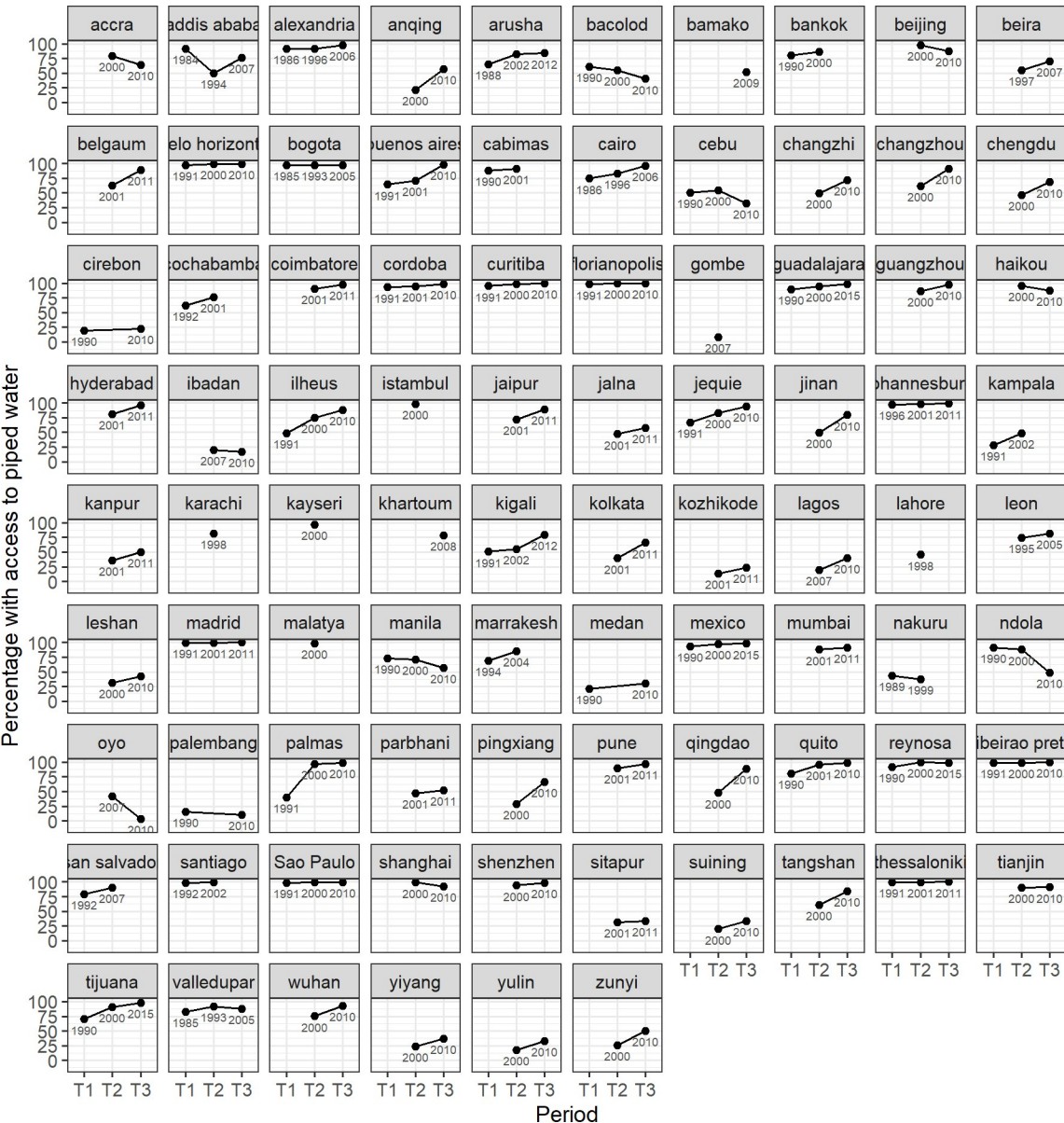

**Fig 7. Percentage of households with access to piped water in 86 cities at different moments in time.**

countries took several days. A fourth limitation is that the possibilities of conducting analysis at the sub-city level are restricted by the level of geographical aggregation available in IPUMS. For some cities, like Mexico City or São Paulo, it is possible to identify dozens of city districts but access to the complete census (instead of a sample) would allow for a much more detailed analysis. This, however, would require a massive storage system and specific statistical procedures to preserve anonymity of the data. Although IPUMS includes only a sample, the tradeoff is that it has solved these logistical issues.

Considering these and other limitations, we do not want to suggest that DUIA is a silver bullet for comparative research. Depending on the questions at hand, DUIA can be useful. While we focused on access to piped water, DUIA also offers the opportunity to study access to e.g. internet or electricity. And while we above studied inequality in terms of households

## Percentage Household with piped water

**Fig 8. Percentage of households with access to piped water by continent and period.**

assets, DUIA also contains data to study inequality in terms of other factors like educational attainment. Perhaps even more promising, it is possible to conduct multivariate analyses, examining, for instance, to what degree educational attainment correlates with household assets or access to amenities. In other front, the access of harmonized multitemporal microdata level opens opportunities in more detailed comparative analysis including typologies of urban development's pathways in association to urban growth analysis. Considering the large number of microdata observations available, robust econometric models can be applied to identify trends and inferences including differences-in-differences-type and propensity-score-matching strategies. In short, DUIA offers a number of additional opportunities relative to other databases.

Although the coverage and reliability of much of the data will continue to depend on painstaking endeavors like taking censuses [28], the processing and analysis of data is greatly enhanced by innovations in data processing, cloud computing, collaborative platforms for code sharing, and open source software. As one of many examples of issues that could be

resolved through technological advances, consider that, at present, we only can disaggregate census data to the administrative units that are used in the census. The incorporation of remote sensing data and machine learning classification for disaggregating census data is certainly one of the most promising developments ahead with already good results such as the Wordpop gridded population [18]. The so-called dasymetric mapping approach they developed could serve as a basis to expand the downscaling of the variables of interest. If integrated with field observations and sampling, these methods could overcome the reliance on administrative units in censuses and integrate remote sensing in urban databases. DUIA offers a wide range of potential future applications. The most obvious is its use in urban benchmarking initiatives such as the monitoring of the Sustainable Development Goals (SDGs). At present, the monitoring of SDGs depends on unharmonized variables, limiting the potential for research over time and especially across space.

## Conclusion

As Acuto and Pejic [29] argue, the global proliferation of city benchmarking initiatives demands critical reflection and constructive engagement from academics. By introducing transparency and replicability in analytical methods, DUIA potentially represents an important initiative to engage collaborative efforts in this direction. DUIA seeks to add to presently available city-level databases by offering opportunities for detailed, replicable, and comprehensive research through extensive metadata, granular primary data, and replicable analysis. The database especially covers cities outside the West, providing new opportunities for comparative research in fields suffering from a dearth of data. We illustrate the potential of DUIA by examining inequality in household assets and access to piped water across a large sample of cities. Although these analyses are exploratory and show marked variation across cities, overall the trends we observe are in the direction of decreasing inequality and greater access.

We acknowledge that DUIA suffers from several limitations. Arguably, the database is not sufficiently detailed for fine-grained case study research and not sufficiently comprehensive for generalizations. Nevertheless, DUIA makes a number of important steps in overcoming obstacles to comparative research. In particular, it offers new opportunities for the examination of broad trends over time through comparatively reliable data and with transparent, replicable procedures.

## Supporting information

**S1 Appendix. City-level databases initiatives.**
(DOCX)

**S2 Appendix. Variables on assets and amenities.**
(TIF)

**S3 Appendix. Delineating cities: The examples of Mexico and Dhaka.**
(DOCX)

## Acknowledgments

The authors wish to acknowledge the statistical offices that provided the underlying data making this research possible: National Institute of Statistics and Censuses, Argentina; National Institute of Statistics, Bolivia; Institute of Geography and Statistics, Brazil; National Institute of Statistics, Chile; National Bureau of Statistics, China; National Administrative Department of Statistics, Colombia; National Institute of Statistics and Censuses, Ecuador; Department of

Statistics and Censuses, El Salvador; Central Agency for Public Mobilization and Statistics, Egypt; Central Statistical Agency, Ethiopia; Bureau of Statistics, Fiji; Ghana Statistical Services, Ghana; National Statistical Office, Greece; Office of the Registrar General & Census Commissioner, India; National Directorate of Statistics and Informatics, Mali; BPS Statistics Indonesia, Indonesia; National Bureau of Statistics, Kenya; National Institute of Statistics, Geography, and Informatics, Mexico; Department of Statistics, Morocco; National Institute of Statistics, Mozambique; National Bureau of Statistics, Nigeria; National Statistics Office, Philippines; National Institute of Statistics, Rwanda; Statistics South Africa, South Africa; National Institute of Statistics, Spain; Central Bureau of Statistics, Sudan; Bureau of Statistics, Tanzania; National Statistical Office, Thailand; Turkish Statistical Institute, Turkey; Bureau of Statistics, Uganda; National Institute of Statistics, Venezuela; and Central Statistics Office, Zambia.

## Author Contributions

**Conceptualization:** Frederico Roman Ramos, Justus Uitermark.

**Data curation:** Frederico Roman Ramos.

**Formal analysis:** Frederico Roman Ramos.

**Methodology:** Frederico Roman Ramos.

**Project administration:** Justus Uitermark.

**Software:** Frederico Roman Ramos.

**Supervision:** Justus Uitermark.

**Writing – original draft:** Frederico Roman Ramos, Justus Uitermark.

**Writing – review & editing:** Frederico Roman Ramos, Justus Uitermark.

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
