## [Decision Letter · Decision Letter 0]

28 Apr 2021

PONE-D-21-09874

Introduction to DUIA: the Database on Urban Inequality and Amenities

PLOS ONE

Dear Dr. Ramos,

Thank you for submitting your manuscript to PLOS ONE. After careful consideration, we feel that it has merit but does not fully meet PLOS ONE’s publication criteria as it currently stands. Therefore, we invite you to submit a revised version of the manuscript that addresses the points raised during the review process.

As highlighted in the Reviewers' Reports, Reviewers 1 and 2 suggest minor changes whereas Reviewer 3 pointed to major issues to be revised. According to Reviewer 3 Report, among others, additional references are required in the introduction section and more explanations on the database would improve the manuscript. Other minor issues are detailed in the Reviewers' Reports.

We look forward to receiving your revised manuscript.

Kind regards,

Eda Ustaoglu, PhD

Academic Editor

PLOS ONE

Journal Requirements:

2. We note that Figure 1 and S3 Appendix in your submission contain map images which may be copyrighted. All PLOS content is published under the Creative Commons Attribution License (CC BY 4.0), which means that the manuscript, images, and Supporting Information files will be freely available online, and any third party is permitted to access, download, copy, distribute, and use these materials in any way, even commercially, with proper attribution. For these reasons, we cannot publish previously copyrighted maps or satellite images created using proprietary data, such as Google software (Google Maps, Street View, and Earth). For more information, see our copyright guidelines: http://journals.plos.org/plosone/s/licenses-and-copyright.

You may seek permission from the original copyright holder of Figure 1 and S3 Appendix to publish the content specifically under the CC BY 4.0 license. 

If you are unable to obtain permission from the original copyright holder to publish these figures under the CC BY 4.0 license or if the copyright holder’s requirements are incompatible with the CC BY 4.0 license, please either i) remove the figure or ii) supply a replacement figure that complies with the CC BY 4.0 license. Please check copyright information on all replacement figures and update the figure caption with source information. If applicable, please specify in the figure caption text when a figure is similar but not identical to the original image and is therefore for illustrative purposes only.

Reviewers' comments:

Reviewer's Responses to Questions

**Comments to the Author**

1. Is the manuscript technically sound, and do the data support the conclusions?

Reviewer #1: Yes

Reviewer #2: Yes

Reviewer #3: Yes

2. Has the statistical analysis been performed appropriately and rigorously? 

Reviewer #1: Yes

Reviewer #2: Yes

Reviewer #3: I Don't Know

3. Have the authors made all data underlying the findings in their manuscript fully available?

Reviewer #1: Yes

Reviewer #2: Yes

Reviewer #3: Yes

4. Is the manuscript presented in an intelligible fashion and written in standard English?

Reviewer #1: Yes

Reviewer #2: Yes

Reviewer #3: Yes

5. Review Comments to the Author

Reviewer #1: A very commendable initiative.

Please address the following minor corrections.

Lines48/49: "and not fully harmonised" should read "and are not fully ..."

Line 204-205 : Please rewrite.

Line 299: "consists in" should read "consists of".

Line 398: "is shows" should read "shows".

Reviewer #2: This work is a clear presentation of what is on the ground.

My point in question 2 above is that the database can be used to show how inequality will be in time to come for the selected cities.

Therefore the analysis can be extended to include projections for the selected cities.

Reviewer #3: This article introduces a new database called the Database on Urban Inequality and Amenities (DUIA), which includes data on the socio-economic development and amenities of 86 cities. DUIA is particularly useful because it provides information on other cities outside of the West and offers opportunities to study asset inequalities. Overall, the paper addresses an important gap in the literature, and the illustrations the authors used were useful. The paper should be considered for publication if the authors address the comments below.

1. In the introduction, the authors raised some important limitations of existing databases (lines 46-58). However, they failed to reference any specific databases or include any citations to support these claims. While some examples are discussed in later sections of the paper, a summary in the introduction will be useful to readers. I recommend that the authors should refer to specific databases as examples in discussing these limitations.

2. Where is the database? The authors described how they extracted and integrated data from the two sources they included in deriving the database. However, they failed to describe the outcome of that process. A section that clearly explains the features of the database will be helpful. This section should be presented before the illustrations.

3. Similar to my comment above, the authors should discuss the application of the database in-depth. The two illustrations are good but are there other applications that you anticipate? I recommend that somewhere, either in the conclusion or limitations, the authors should clearly discuss the nature of uses they anticipate for their database.

4. The authors should check for word omissions and punctuation issues. A few of them are listed below

Line 38-39: there is an “of” missing in … proliferation databases. Also, there is an extra “to” after to go.

Line 64-65: a “to” is missing

Line 74-75: delete study

Punctuations: e.g., lines 169, 166

6. PLOS authors have the option to publish the peer review history of their article (what does this mean?). If published, this will include your full peer review and any attached files.

Reviewer #1: No

Reviewer #2: No

Reviewer #3: No

---

## [Author Response · Author response to Decision Letter 0]

23 May 2021

Academic editor:

We have carefully checked the templates and made the adjustments to meet the journal requirements. 

2. We note that Figure 1 and S3 Appendix in your submission contain map images which may be copyrighted. All PLOS content is published under the Creative Commons Attribution License (CC BY 4.0).

All figures were produced by the authors specifically for this manuscript and no copyrighted images were used in their production. 

3. Please include captions for your Supporting Information files at the end of your manuscript, and update any in-text citations to match accordingly. 

We have checked the templates and made the adjustments to meet the journal requirements. 

Reviewer #1:

4. Please address the following minor corrections.

We have addressed all the indicated minor corrections accordingly. 

Reviewer #2:

5. My point is that the database can be used to show how inequality will be in time to come for the selected cities. Therefore, the analysis can be extended to include projections for the selected cities.

We have included a paragraph in the section ‘Limitations and opportunities ahead’ to discuss possible applications, including econometric analysis using microdata level. Nevertheless, we are skeptical about the potential to conduct robust projections mainly due to the sparse time data availability. 

Reviewer #3:

6. In the introduction, the authors raised some important limitations of existing databases (lines 46-58). However, they failed to reference any specific databases or include any citations to support these claims. While some examples are discussed in later sections of the paper, a summary in the introduction will be useful to readers. I recommend that the authors should refer to specific databases as examples in discussing these limitations.

We have included the references in the introductory section as suggested. 

7. Where is the database? The authors described how they extracted and integrated data from the two sources they included in deriving the database. However, they failed to describe the outcome of that process. A section that clearly explains the features of the database will be helpful. This section should be presented before the illustrations.

We agree and incorporated a new subsection in the third section (before the illustrations) describing the components and features of the database. This new subsection highlights the use of a Github repository for the R codes that enables replication and the use of a Figshare webpage to distribute the aggregated-level database. We will provide and include the links and respective DOIs after passing the point of anonymous peer review.

8. Similar to my comment above, the authors should discuss the application of the database in-depth. The two illustrations are good but are there other applications that you anticipate? I recommend that somewhere, either in the conclusion or limitations, the authors should clearly discuss the nature of uses they anticipate for their database.

As responded to reviewer #2, we included a paragraph in the section ‘Limitations and opportunities ahead’ dedicated to possible applications. These applications include multivariate statistical analysis, logistical regressions, and comparative analysis.

---

## [Decision Letter · Decision Letter 1]

14 Jun 2021

An introduction to DUIA: the Database on Urban Inequality and Amenities

PONE-D-21-09874R1

Dear Dr. Ramos,

We’re pleased to inform you that your manuscript has been judged scientifically suitable for publication and will be formally accepted for publication once it meets all outstanding technical requirements.

Kind regards,

Eda Ustaoglu, PhD

Academic Editor

PLOS ONE

Additional Editor Comments (optional):

Reviewers' comments:

Reviewer's Responses to Questions

**Comments to the Author**

1. If the authors have adequately addressed your comments raised in a previous round of review and you feel that this manuscript is now acceptable for publication, you may indicate that here to bypass the “Comments to the Author” section, enter your conflict of interest statement in the “Confidential to Editor” section, and submit your "Accept" recommendation.

Reviewer #1: All comments have been addressed

Reviewer #2: All comments have been addressed

Reviewer #3: All comments have been addressed

2. Is the manuscript technically sound, and do the data support the conclusions?

Reviewer #1: Yes

Reviewer #2: Yes

Reviewer #3: Yes

3. Has the statistical analysis been performed appropriately and rigorously? 

Reviewer #1: Yes

Reviewer #2: Yes

Reviewer #3: Yes

4. Have the authors made all data underlying the findings in their manuscript fully available?

Reviewer #1: Yes

Reviewer #2: Yes

Reviewer #3: Yes

5. Is the manuscript presented in an intelligible fashion and written in standard English?

Reviewer #1: Yes

Reviewer #2: Yes

Reviewer #3: Yes

6. Review Comments to the Author

Reviewer #1: The authors should be congratulated for a very useful initiative. I have no further comments to make.

Reviewer #2: The comments and suggestions have been added to the work. It has useful illustrations and a well summarized abstract.

In general, it should be considered for publication.

Reviewer #3: (No Response)

7. PLOS authors have the option to publish the peer review history of their article (what does this mean?). If published, this will include your full peer review and any attached files.

Reviewer #1: No

Reviewer #2: No

Reviewer #3: No

---

## [Editor Report · Acceptance letter]

17 Jun 2021

PONE-D-21-09874R1 

An introduction to DUIA: the database on urban inequality and amenities 

Dear Dr. Ramos:

I'm pleased to inform you that your manuscript has been deemed suitable for publication in PLOS ONE. Congratulations! Your manuscript is now with our production department. 

Kind regards, 

on behalf of

Dr. Eda Ustaoglu 

Academic Editor

PLOS ONE